# The NKCC1 antagonist bumetanide mitigates interneuronopathy associated with ethanol exposure in utero

Alexander GJ Skorput[1,2†], Stephanie M Lee[1†], Pamela WL Yeh[1], Hermes H Yeh[1]*

[1]Department of Molecular and Systems Biology, Geisel School of Medicine at Dartmouth, Hanover, United States; [2]Department of Neuroscience, School of Medicine, University of Minnesota Twin Cities, Minneapolis, United States

**Abstract** Prenatal exposure to ethanol induces aberrant tangential migration of corticopetal GABAergic interneurons, and long-term alterations in the form and function of the prefrontal cortex. We have hypothesized that interneuronopathy contributes significantly to the pathoetiology of fetal alcohol spectrum disorders (FASD). Activity-dependent tangential migration of GABAergic cortical neurons is driven by depolarizing responses to ambient GABA present in the cortical enclave. We found that ethanol exposure potentiates the depolarizing action of GABA in GABAergic cortical interneurons of the embryonic mouse brain. Pharmacological antagonism of the cotransporter NKCC1 mitigated ethanol-induced potentiation of GABA depolarization and prevented aberrant patterns of tangential migration induced by ethanol in vitro. In a model of FASD, maternal bumetanide treatment prevented interneuronopathy in the prefrontal cortex of ethanol exposed offspring, including deficits in behavioral flexibility. These findings position interneuronopathy as a mechanism of FASD symptomatology, and posit NKCC1 as a pharmacological target for the management of FASD.

DOI: https://doi.org/10.7554/eLife.48648.001

*For correspondence:
hermes.yeh@dartmouth.edu

†These authors contributed equally to this work

**Competing interests:** The authors declare that no competing interests exist.

## Introduction

Binge-type exposure of the embryonic mouse brain to a moderate level of ethanol leads to interneuronopathy, hallmarked by aberrant tangential migration of corticopetal GABAergic interneurons derived from the medial ganglionic eminence (MGE) (*Skorput et al., 2015*). Interneuronopathies are widely implicated in neurodevelopmental disorders (*Catterall, 2018*; *Kato and Dobyns, 2005*; *Katsarou et al., 2017*), including Fetal Alcohol Spectrum Disorders (FASD) (*Skorput et al., 2015*; *Skorput and Yeh, 2016*). Data from the Behavioral Risk Factor Surveillance System for the years 2015–2017 show self-reported rates of current drinking, and binge drinking ($\geq$4 drinks) within the past 30 days to be 11.5% and 3.9%, respectively, among pregnant woman aged 18–44 years. Pregnant binge drinkers averaged 4.5 binge episodes in the past 30 days with an average binge intensity of 6.0 drinks (*Denny et al., 2019*). As a result of such binge drinking episodes, FASD remains a major public health concern in the United States, and a recent assessment of first-graders in 4 US communities conservatively estimated the prevalence rates of FASD to range from 1.1% to 5.0% (*May et al., 2018*). The global prevalence rate of FASD is estimated at 7.7 per 1000 children, and is as high as 111.1 per 1000 children among some populations (*Lange et al., 2017*). Despite this prevalence, our understanding of the embryonic pathoetiology of FASD is incomplete, hampering development of targeted medication and broadly applicable treatment strategies.

The more efficacious current treatments for FASD rely primarily on postnatal behavioral therapy (*Nash et al., 2017*), or pharmaceutics aimed to treat the symptoms, rather than the etiology, of this neurodevelopmental disorder (*Peadon et al., 2009*; *Rowles and Findling, 2010*). Indeed, there is a

societal need to uncover the cellular and subcellular underpinnings by which ethanol adversely affects neurodevelopmental processes (*Ismail et al., 2010*; *Kodituwakku, 2010*; *Pruett et al., 2013*). Here, we report on pharmacologic mitigation of ethanol-induced interneuronopathy and associated executive function deficits in a preclinical model of FASD.

GABAergic neuroblasts arising in the MGE are fated to become parvalbumin ($PV^+$) or somatostatin expressing inhibitory interneurons in the cortex. Approximately 50% of this MGE-derived population will mature to become $PV^+$ fast-spiking basket cells which are electrically coupled across the deep layers of the neocortex and pace oscillatory gamma rhythms, which are required for higher cognition (*Batista-Brito and Fishell, 2009*; *Xu et al., 2008*). Thus, the corticopetal tangential migration and maturation of MGE-derived GABAergic interneurons is critical to the establishment of inhibitory/excitatory balance in intracortical circuits, and proper cortical functioning (*Batista-Brito and Fishell, 2009*; *Catterall, 2018*).

Building on our previous work (*Cuzon et al., 2008*; *Cuzon et al., 2006*), we hypothesized that ethanol's positive allosteric modulation of $GABA_A$ receptors ($GABA_AR$) expressed on migrating neuroblasts is the mechanism by which in utero ethanol exposure causes enduring interneuronopathy in models of FASD (*Skorput et al., 2015*; *Skorput and Yeh, 2016*). In these immature neurons, $GABA_AR$ activation results in membrane depolarization (*Ben-Ari, 2014*; *Ben-Ari, 2002*; *Owens and Kriegstein, 2002*). This paradoxical depolarizing action has been postulated to drive the corticopetal tangential migration of GABAergic interneurons during corticogenesis (*Behar et al., 1996*; *Ben-Ari, 2002*; *Ben-Ari et al., 2012*; *Cuzon et al., 2006*; *Wang and Kriegstein, 2009*). To test the influence of GABAergic depolarization on ethanol-induced interneuronopathy, we sought to normalize ethanol's potentiation of GABAergic depolarization by reducing the driving force of GABA-induced membrane depolarization via a decrease in the intracellular concentration of the $GABA_AR$ permeable anion $Cl^-$ ($[Cl^-]_i$).

To this end, we targeted the chloride importing $Na^+$-$K^+$−$2Cl^-$ isoform one cotransporter (NKCC1) because it is expressed at high levels in the embryonic brain when its activity predominates over that of the chloride exporting $K^+$-$Cl^-$ co-transporter (KCC2). This differential activity results in a net higher $[Cl^-]_i$ compared to $[Cl^-]_o$. As such, the reversal potential for chloride, and that of GABA-activated responses ($E_{GABA}$), is set at a membrane potential that is depolarized relative to the resting membrane potential. Thus, $GABA_AR$ activation causes chloride extrusion and membrane depolarization in these migrating neuroblasts (*Ben-Ari, 2014*; *Ben-Ari, 2002*; *Owens and Kriegstein, 2002*). The loop diuretic bumetanide is an NKCC1 antagonist that shifts the $E_{GABA}$ of embryonic neuroblasts to a more negative membrane potential when administered maternally (*Wang and Kriegstein, 2011*). Given these considerations, we tested the hypothesis that in utero treatment with bumetanide during the period of prenatal ethanol exposure will mitigate manifestation of prenatal ethanol-induced aberrant tangential migration, and prevent deficits in behavioral flexibility.

The timing of gestational exposure to ethanol is a key determinant of the diagnostic outcome of FASD (*May et al., 2013*; *Pruett et al., 2013*). Corticogenesis occurs in earnest during the mid-first trimester when the developing brain is highly vulnerable to insult by ethanol (*Ayoola et al., 2009*; *Clancy et al., 2001*; *May et al., 2014*). We established a mouse model that simulates an early gestational, mid-first trimester human equivalent, exposure to binge-type maternal ethanol consumption from embryonic day (E) 13.5 - E16.5 (*Clancy et al., 2001*; *Skorput et al., 2015*). Using this model, we demonstrated enhanced entry of MGE-derived GABAergic interneurons into the prefrontal cortex (PFC), a persistent increase in the number of $PV^+$ interneurons in the young adult PFC, and impairment in the PFC-dependent behavioral flexibility of offspring (*Skorput et al., 2015*).

In the present study, we used this mouse model to assess whether the NKCC1 cotransporter is a tractable pharmacological target for normalizing in utero ethanol exposure-induced escalation of tangential migration to the prefrontal cortex. We report here that ethanol induces a shift in $E_{GABA}$ toward a more depolarized potential, accounting at least in part for ethanol's potentiation of GABA-induced depolarizing responses in embryonic MGE-derived GABAergic interneurons. In the short term, antagonizing NKCC1 with bumetanide mitigated ethanol-induced aberrant tangential migration. In the long term, bumetanide treatment in vivo prevented prenatal ethanol exposure-induced interneuronopathy in the prefrontal cortex, and the associated deficits in PFC-dependent behavioral flexibility. Our findings support the feasibility of a pharmacological strategy to target NKCC1 for the management of FASD.

## Results

### Ethanol induces a depolarizing shift in the GABA reversal potential of embryonic MGE-derived GABAergic cortical interneurons that is normalized by the NKCC1 inhibitor bumetanide

Nkx2.1$^+$ embryonic MGE-derived GABAergic cortical interneurons were identified by tdTomato fluorescence and targeted for perforated patch clamp recording in acute (200 µm) telencephalic slices obtained from E14.5–16.5 Nkx2.1Cre/Ai14 mouse brain (*Figure 1a*). GABA (50 µM), ethanol (6.5 mM) and aCSF were loaded into separate barrels of a multi-barrel drug pipette and focally applied by regulated pressure either individually, or in combination, in the immediate vicinity of the cell under study (*Figure 1a*). Since GABA-induced membrane depolarization is prevalent in many types of immature neurons early in brain development and has been postulated to promote the tangential migration of GABAergic cortical interneurons (*Behar et al., 1996*; *Ben-Ari, 2002*; *Ben-Ari et al., 2012*; *Cuzon et al., 2006*; *Wang and Kriegstein, 2009*), we asked whether exogenously applied GABA depolarized embryonic Nkx2.1$^+$ cortical interneurons. In *Figure 1b*, the representative sets of digitized raw traces display current responses to 500 ms pressure applications of 50 µM GABA monitored at varying holding potentials before (black traces) and during ethanol exposure (red traces). The polarity and peak amplitude of each GABA response were plotted as a function of the corresponding holding potential, and the intercept of the current-voltage plot along the abscissa was used to estimate $E_{GABA}$. *Figure 1c* shows linear regression of the group mean data. The control mean $E_{GABA}$ (dotted black line in *Figure 1c*) was −23.18 ± 1.45 mV, which was significantly more depolarized than the mean resting membrane potential, estimated in whole-cell mode (dotted blue line in *Figure 1c*; −41.9 ± 2.3 mV; unpaired t-test, p<0.01). Importantly, *Figure 1c* illustrates that ethanol exposure shifted $E_{GABA}$ rightward relative to that assessed during the control epoch (dotted red line) and subsequent washout of ethanol with aCSF shifted $E_{GABA}$ back to the control level (dotted gray line). *Figure 1d* illustrates the mean change in $E_{GABA}$ for each litter before and during ethanol exposure, indicating that the mean $E_{GABA}$ of control cells was shifted to significantly more depolarized membrane potentials with exposure to 6.5 mM ethanol (*Figure 1d*; −16.52 ± 0.93 mV; p = 0.0077) and shifted back to the control level following washout of ethanol with aCSF (gray dots; −21.57 ± 2.51 mV; p = 0.81; effect size $\omega^2$ = 0.17). The finding that ethanol exposure results in a depolarizing shift in $E_{GABA}$ suggests that it increases $[Cl^-]_i$ and thus, chloride drive. Exposure to 50 mM ethanol, the concentration used to treat organotypic slice cultures (Figure 3), also exerted a depolarizing shift in $E_{GABA}$ (*Figure 1—figure supplement 1*; aCSF; −37.36 ± 1.5 mV; EtOH; −22.15 ± 1.1 mV; n = 6 litters, 21 cells total; p = 0.0003; effect size d = 4.58).

Following a 30 min pretreatment of acute E14.5–16.5 telencephalic slices with the NKCC1 antagonist bumetanide (20 µM), ethanol still shifted $E_{GABA}$ rightward relative to control (*Figure 1—figure supplement 2*). However, bumetanide pretreatment normalized the ethanol-induced depolarization of $E_{GABA}$ to control levels (*Figure 1e*; EtOH+Bumet; −24.82 ± 2.52 mV; n = 5 litters, 13 cells total; p = 0.56; effect size $\omega^2$ = 0.55). The basis of this normalization is likely a functional antagonism as bumetanide pretreatment significantly hyperpolarized $E_{GABA}$ in Nkx2.1$^+$ GABAergic interneurons compared to controls (*Figure 1e*; Bumet; −34.17 4.02 mV; n = 5 litters, 13 cells total; p = 0.004), and the magnitude of the depolarizing shift in $E_{GABA}$ induced by exposure to 6.5 mM ethanol was not effected by bumetanide pretreatment (*Figure 1f*; Control; 27.08 ± 5.61 mV; Bumet pretreatment; −25.38 ± 6.41 mV; p = 0.37; effect size d = 0.11). These results prompted us to ask whether bumetanide treatment may restrict the enhanced chloride driving force caused by ethanol-induced increases in $[Cl^-]_i$ to thus mitigate ethanol's potentiation of depolarizing GABA$_A$R activity in embryonic GABAergic interneurons.

### Bumetanide attenuates ethanol-induced potentiation of depolarizing GABA responses in embryonic MGE-derived GABAergic cortical interneurons

In perforated patch-clamp recordings, we analyzed current response amplitudes of GABAergic cortical interneurons held at −60 mV to 500 ms applications of 50 µM GABA before and during 6.5 mM ethanol exposure in E14.5-E16.5 telencephalic slices (*Figure 2a*, inset). *Figure 2a* illustrates that, relative to baseline conditions, the mean GABA response amplitude during ethanol exposure is

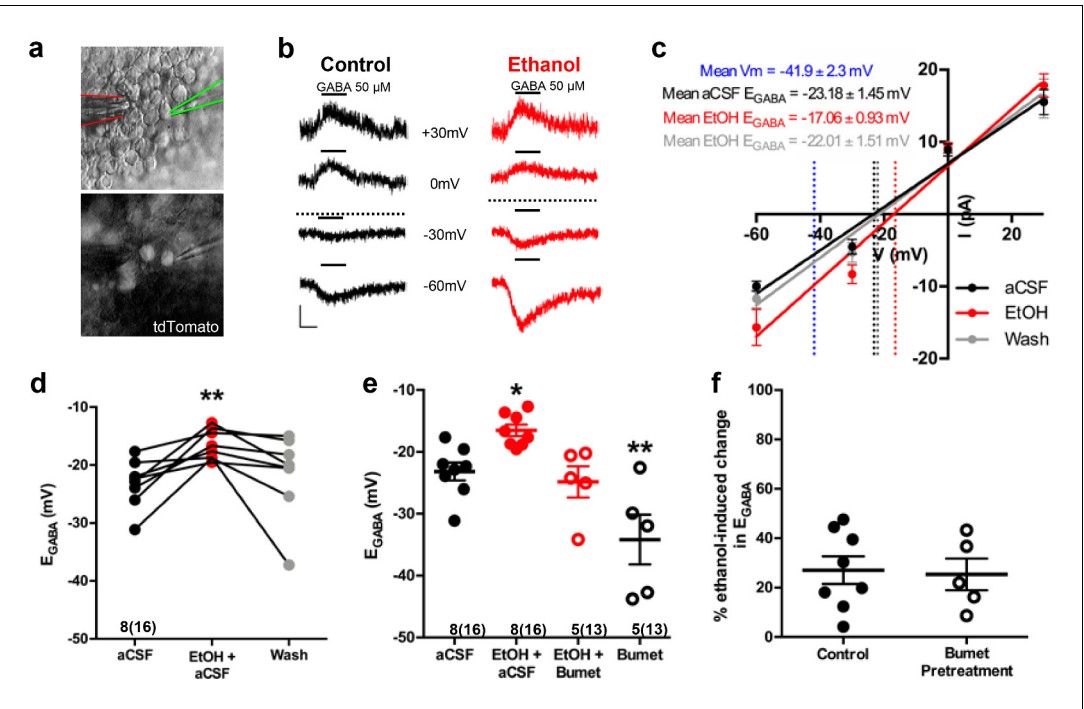

**Figure 1.** Ethanol induces a depolarizing shift in GABA reversal potential in embryonic MGE-derived GABAergic cortical interneurons that is normalized by the NKCC1 inhibitor bumetanide. (a) Hoffman modulated contrast image of acute telencephalic slice of E14.5 mouse brain showing patch clamp electrode (green outline) holding a tdTomato fluorescent Nkx2.1[+] MGE-derived GABAergic interneuron in the perforated-patch voltage clamp configuration, with a multi-barrel drug pipette (red outline) positioned in the vicinity. (b) $E_{GABA}$ was empirically determined by focal application of GABA (black bars, 50 μM) at varying holding potentials under control (black traces), and ethanol exposure (6.5 mM EtOH, red traces), conditions. Dotted lines indicate $E_{GABA}$ for each condition, scale bar vertical = 10 pA; horizontal = 500 ms. (c) I/V plot of peak GABA-induced current over holding potential defines $E_{GABA}$ as the x-intercept under control (black), ethanol exposure (red), and aCSF wash (gray) conditions. Dotted blue line denotes mean resting membrane potential. (d) $E_{GABA}$ of neurons under control conditions (aCSF), during ethanol exposure (EtOH+aCSF), and aCSF wash (Wash). ** = p < 0.01 one-way ANOVA with Dunnett post-test. (e) Group-wise comparison of $E_{GABA}$ in Nkx2.1[+] neurons without and with bumetanide pretreatment (Bumet, 20 μM) and during exposure to EtOH with bumetanide pretreatment (EtOH+Bumet). Numbers below scatter denote number of litters and number of cells recorded in (). *, ** = p < 0.05, p < 0.01 compared to aCSF, one-way ANOVA with Dunnett post-test. (f) Magnitude of change in $E_{GABA}$ induced by 6.5 mM ethanol without and with bumetanide pretreatment (p > 0.05, unpaired t-test).

DOI: https://doi.org/10.7554/eLife.48648.002

The following figure supplements are available for figure 1:

**Figure supplement 1.** 50 mM ethanol induces a depolarizing shift in GABA reversal potential in embryonic MGE-derived GABAergic cortical interneurons.

DOI: https://doi.org/10.7554/eLife.48648.003

**Figure supplement 2.** I/V plot of peak GABA-induced current over holding potential in bumetanide pre-treated cells.

DOI: https://doi.org/10.7554/eLife.48648.004

significantly greater (aCSF; 10.02 ± 0.99 pA; EtOH+aCSF; 16.89 ± 2.39 pA; n = 8 litters, 17 cells total; p = 0.0039; effect size $\omega^2$ = 0.058) and the subsequent washout of ethanol with aCSF reverses the effect of ethanol on GABA response amplitude (Washout; 12.60 ± 1.99 pA; p = 0.16). The mean GABA response amplitude during 50 mM ethanol exposure was also significantly greater than that of control (*Figure 2—figure supplement 1*; aCSF; 7.77 ± 1.09 pA; EtOH; 17.28 ± 3.21 pA; n = 6 litters, 21 cells total; p =0.01). The mean amplitude of the GABA-activated current responses of GABAergic interneurons in the bumetanide pretreated slices was 10.53 ± 0.62 pA before ethanol exposure, and was significantly increased to 13.45 ± 1.31 pA during ethanol exposure (*Figure 2b*;

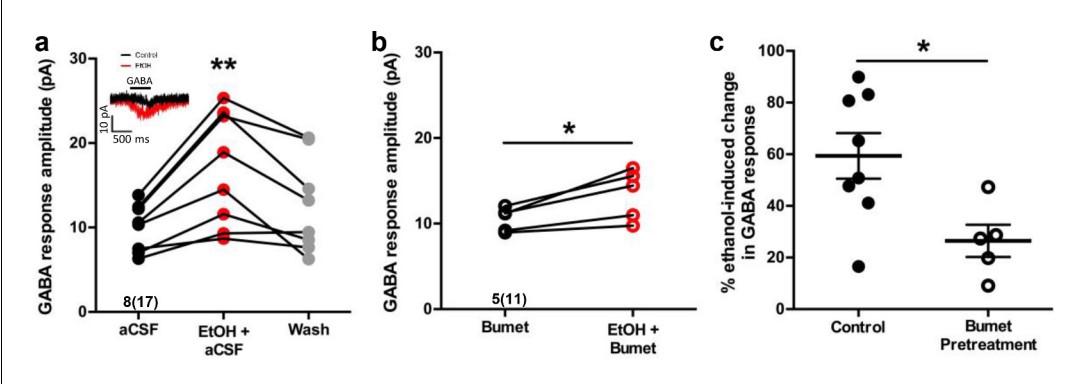

**Figure 2.** Bumetanide attenuates ethanol-induced potentiation of depolarizing GABA responses in embryonic MGE-derived GABAergic cortical interneurons. (a) Peak current amplitude recorded from Nkx2.1+ MGE-derived GABAergic interneurons in slices of E14.5 mouse telencephalon in response to focal application of GABA (50 µM) under control (aCSF; inset: black trace), ethanol exposure (EtOH, 6.5 mM; inset: red trace), or aCSF wash (Wash), scale bar vertical = 10 pA; horizontal = 500 ms. ** = p < 0.01 one-way ANOVA with Dunnett post-test. (b) Peak current amplitude in response to focal application of GABA (50 µM) in the presence of bumetanide (Bumet) or ethanol and bumetanide (EtOH+Bumet). * = p < 0.01 paired t-test. (c) Percent change in peak GABA response induced by ethanol exposure at baseline (aCSF) and in the presence of bumetanide (Bumet). * = p < 0.05 unpaired t-test.

DOI: https://doi.org/10.7554/eLife.48648.005

The following figure supplements are available for figure 2:

**Figure supplement 1.** 50 mM ethanol induces a potentiation of depolarizing GABA responses in embryonic MGE-derived GABAergic interneurons.

DOI: https://doi.org/10.7554/eLife.48648.006

**Figure supplement 2.** Bumetanide treatment does not change the GABA response amplitude.

DOI: https://doi.org/10.7554/eLife.48648.007

n = 5 litters, recorded from 11 cells; p = 0.019; effect size d = 1.71). Bumetanide pretreatment alone did not alter the GABA response amplitude compared to aCSF control (*Figure 2—figure supplement 2*; p = 0.71). However, ethanol exposure potentiated the amplitude of GABA responses by 59.37 ± 8.83% under control conditions, and by 26.46 ± 6.26% in the bumetanide pretreatment group, demonstrating the ability of bumetanide to significantly decrease the mean ethanol-induced change in GABA response (*Figure 2c*; p = 0.011; effect size d = 1.63). Taken together with our earlier finding that GABA-activated current responses in embryonic MGE-derived GABAergic interneurons are mediated by GABA$_A$ receptors (*Cuzon Carlson and Yeh, 2011*; *Cuzon et al., 2008*), these results indicate that NKCC1 antagonism attenuates ethanol-induced potentiation of depolarizing GABA$_A$R-activated responses in migrating GABAergic cortical interneurons.

## Co-treatment with bumetanide prevents ethanol's enhancement of tangential migration in vitro

We then asked whether NKCC1 antagonism inhibits ethanol-induced escalation of tangential migration among cortical GABAergic interneurons (*Cuzon et al., 2008*; *Skorput et al., 2015*; *Skorput and Yeh, 2016*). To this end, organotypic slice cultures containing the embryonic PFC were prepared from E14.5 Nkx2.1Cre/Ai14 brains, incubated in control or ethanol-containing medium without or with bumetanide, and tangential migration of MGE-derived interneurons was assessed following 24 hr in culture (*Figure 3*). Nkx2.1+ GABAergic interneurons were counted in consecutive bins 100 µm in width arranged ventro-dorsally and beginning at the cortico-striate junction (*Figure 3a*). Relative to controls, organotypic telencephalic slices cultured in media containing 50 mM ethanol had more Nkx2.1+ GABAergic interneurons per 100 µm cortical bin in the embryonic PFC (*Figure 3a and b*; Control $\bar{x}$ = 61.3 ± 2.8 cells, 12 cultures; EtOH $\bar{x}$ = 78.5 ± 3.5 cultures; P = 0.00166; effect size $\omega^2$ = 0.31). The addition of bumetanide (20µM) to the culture medium prevented the ethanol-induced escalation of Nkx2.1+ interneuron migration into the cortex with no significant difference observed compared to control (*Figure 3a and b*; EtOH+Bumet $\bar{x}$ = 61.1 ± 2.4 cells, 10 cultures; P > 0.999; effect size $\omega^2$ for treatment = 0.23). Bumetanide alone (20 µM) did not

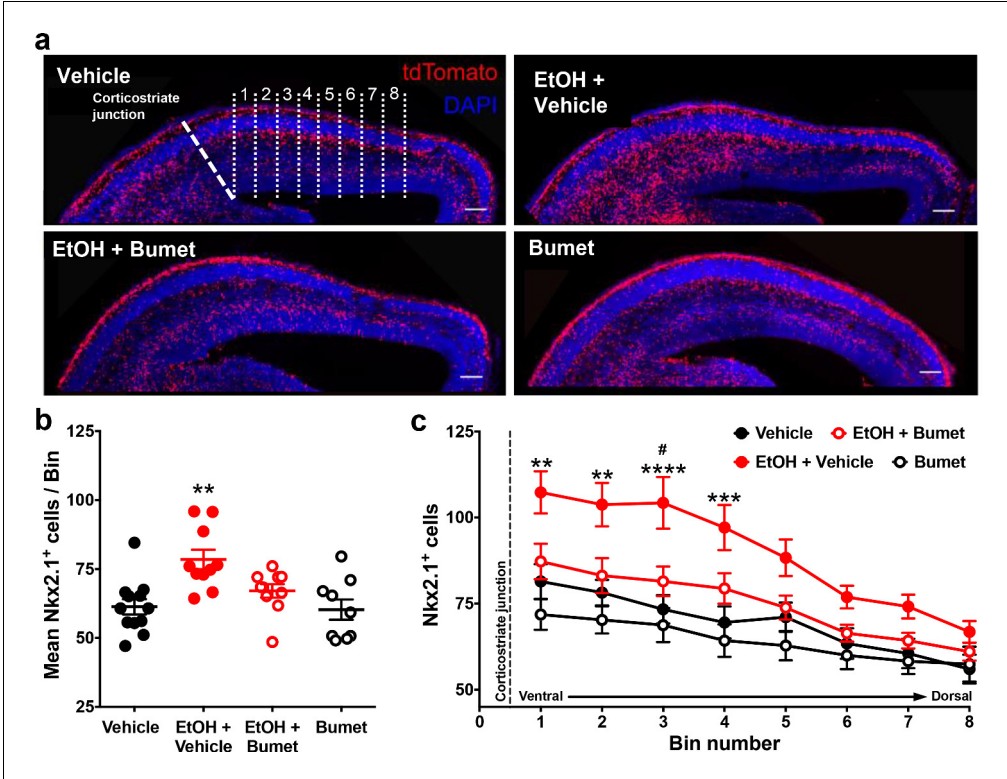

**Figure 3.** Co-treatment with bumetanide prevents ethanol's enhancement of tangential migration in vitro. (a) Fluorescent images of organotypic E14.5 Nkx2.1-Cre/Ai14 mouse telencephalic brain slices treated with vehicle, ethanol (50 mM) + vehicle (EtOH + Vehicle), ethanol + bumetanide (EtOH+Bumet, 20 µM), or bumetanide alone (Bumet, 20 µM), and counterstained with DAPI (blue). The corticostriate junction is marked by the dashed line, and numbered counting bins are denoted by dotted lines in the vehicle panel. Scale bars = 100 µm. (b) Mean number of tdTomato flourecent Nkx2.1+ cells per 100 µm bin. ** = p < 0.01 one-way ANOVA with Bonferroni post-test. Each dot represents one organotypic culture. (c) Number of tdTomato fluorescent Nkx2.1+ cells in each 100 µm bin. **, ***, **** = P < 0.01, 0.001, 0.0001 compared to control, # = p < 0.05 compared to EtOH+Bumet two-way ANOVA with Bonferroni post-test.

DOI: https://doi.org/10.7554/eLife.48648.008

affect Nkx2.1+ interneuron entry into the cortex compared to controls (*Figure 3a and b*; Bumet $\bar{x}$ = 60.3 ± 3.7 cells, 9 cultures; P > 0.999). When analyzed in terms of the number of Nkx2.1+ cells per cortical bin, the largest increase in Nkx2.1+ cell number was in the cortical region most proximal to the corticostriate juncture (*Figure 3c*; effect size $\omega^2$ for bin = 0.23). These results are consistent with aberrant tangential migration being a primary effect of ethanol exposure in the fetal brain. In addition, these data demonstrate that bumetanide can inhibit the ethanol-induced supranormal cortico-petal migration of GABAergic interneurons at concentrations that did not affect their normal pattern of migration.

## Maternal bumetanide treatment prevents ethanol-induced escalation of tangential migration in vivo

To assess the short term interaction between binge ethanol exposure and bumetanide treatment in vivo, we co-treated binge-type ethanol-consuming (5% EtOH w/w) pregnant dams harboring Nkx2.1Cre/Ai14 embryos daily with bumetanide (i.p.; 0.15mg/kg dissolved in DMSO) from E13.5 – 16.5 and analyzed the density of Nkx2.1+ profiles in the embryonic PFC at E16.5 (*Figure 4a*). Ethanol exposure alone significantly increased the number of Nxk2.1+ neurons in the dorsomedial telencephalon (*Figure 4b*; Control $\bar{x}$ = 1.78x10$^{-3}$ ± 5.90x10$^{-5}$ cells/µm$^2$, 10 brains from 3 litters; EtOH $\bar{x}$ = 2.61x10$^{-3}$ ± 1.39x10$^{-4}$ cells/µm$^2$, 10 brains from 4 litters; P = 1.13x10$^{-5}$; effect size $\omega^2$= 0.65). Maternal treatment over the course of binge-type ethanol exposure with bumetanide significantly

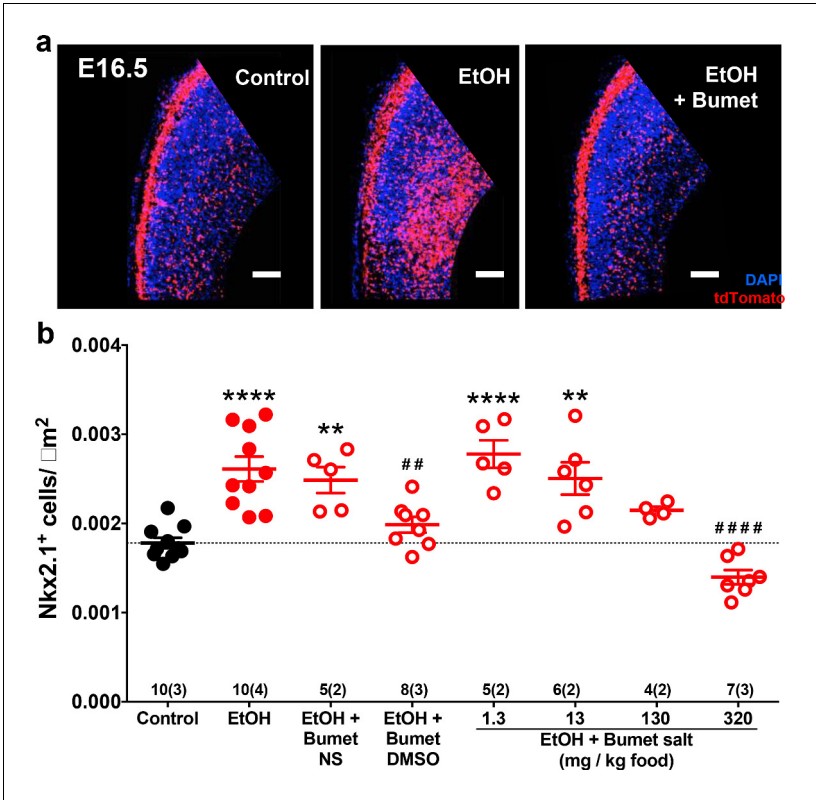

**Figure 4.** Maternal bumetanide treatment prevents ethanol-induced escalation of tangential migration in vivo. Nkx2.1[+] cells were identified by tdTomato fluorescence and quantified in the dorso-medial telencephalon of E16.5 embryonic brain. (a) Fluorescent images counterstained with DAPI following control, binge-type maternal ethanol consumption from E13.3 - E16.5 (EtOH), or ethanol exposure in combination with maternal bumetanide treatment administered i.p. 0.15 mg/kg/day (E13.5 - E16.5) dissolved in DMSO (EtOH+Bumet). Scale bars = 100 μm. (b) Quantification of Nkx2.1[+] cells in the dorso-medial telencephalon after ethanol exposure in combination with maternal bumetanide treatment administered i.p. 0.15 mg/kg/day (E13.5 - E16.5) dissolved in normal saline (EtOH +Bumet NS) or DMSO (EtOH+Bumet DMSO), or maternal bumetanide treatment administered in the ethanol containing liquid diet (EtOH+Bumet salt; 1.3, 13, 130, 320 mg/kg food), or control conditions under which dams consumed equicaloric liquid food from E13.5-E16.5 (control). Dotted line highlights mean of control group. Numbers above x-axis denote number of brains and number of litters in (). **,**** = p < 0.01, 0.0001 compared to control; ##, #### = p < 0.01, 0.0001 compared to EtOH; one-way ANOVA with Bonferroni post-test. Control and EtOH groups are reproduced from *Skorput et al. (2015)*.
DOI: https://doi.org/10.7554/eLife.48648.009

attenuated the ethanol-induced increase in MGE-derived GABAergic interneuron density in the embryonic PFC (*Figure 4b*; EtOH+Bumet DMSO i.p. $\bar{x}$ = 1.99x10$^{-3}$ ± 8.79x10$^{-5}$ cells/μm$^2$, 8 brains from 3 litters; P = 0.00357). In addition, bumetanide co-treatment completely normalized the ethanol-induced effect, returning MGE-derived interneuron density to control levels (*Figure 4b*; P >0.999). As a negative control, given that bumetanide is relatively insoluble in aqueous solutions, treatment with normal saline to which bumetanide (0.015 mg/ml) was added failed to decrease the ethanol induced enhancement of tangential migration when injected i.p. (*Figure 4b*; EtOH+Bumet NS $\bar{x}$ = 2.49x10$^{-3}$ ± 1.45x10$^{-5}$ cells/μm$^2$, 5 brains from 2 litters; P >0.999), and a significant increase in Nkx2.1[+] cell density persisted in the E16.5 PFC compared to controls (*Figure 4b*; P = 0.00481).

We next sought to simulate a more clinically relevant route of bumetanide administration. We asked whether bumetanide, administered orally in the course of ad lib feeding, can prevent the in vivo effects of binge-type ethanol exposure on the entry of MGE-derived interneurons to the embryonic PFC. To this end, a Na[+] bumetanide salt (320, 130, 13, 1.3 mg/kg food) was dissolved in the 5% w/w ethanol containing liquid diet fed to binge ethanol-consuming dams.

E16.5 embryos from dams ingesting ethanol-containing liquid food with 320 mg bumetanide salt /kg food had a significantly lower density of Nkx2.1$^+$ cells in the PFC compared to ethanol exposure alone (Fig 4b; EtOH+Bumet 320mg/kg food $\bar{x}$ = 1.40x10$^{-3}$ ± 7.96x10$^{-5}$ cells/μm$^2$, 7 brains from 2 litters; P = 1.38x10$^{-8}$) with complete normalization relative to the control cohort (*Figure 4b*; P = 0.483). At a concentration of 130 mg bumetanide salt/kg food, no significant difference in Nkx2.1$^+$ cell density was found in the E16.5 PFC compared to control or binge ethanol exposure alone (*Figure 4*; EtOH+Bumet 130mg/kg food $\bar{x}$ = 2.15x10$^{-3}$ ± 4.07x10$^{-5}$ cells/μm$^2$, 4 brains from 2 litters; P > 0.999, P = 0.465 respectively). Embryos born to dams that consumed a liquid diet containing both 5% w/w ethanol and the lowest concentrations of bumetanide tested (1.3 or 13mg bumetanide salt / kg food) showed significantly higher densities of MGE-derived GABAergic interneurons in the PFC compared to controls (*Figure 4b*; EtOH+Bumet 1.3mg/kg food $\bar{x}$ = 2.78x10$^{-3}$ ± 1.55x10$^{-4}$ cells/μm$^2$, 5 brains from 2 litters; EtOH+Bumet 13mg/kg food $\bar{x}$ = 2.50x10$^{-3}$ ± 1.81 x 10$^{-4}$ cells/μm$^2$, 6 brains from 2 litters; P = 1.65x10$^{-5}$, P = 0.00152, respectively) and Nkx2.1$^+$ cell densities were not significantly decreased compared to EtOH exposure alone (*Figure 4b*; P > 0.999). DMSO alone administered i.p. was not performed as a control for this analysis since bumetanide administration via a different vehicle (bumetanide salt in food) produced similar results. In addition, bumetanide in normal saline delivered i.p. controlled for the i.p. route of administration. Bumetanide alone had no effect in vitro (*Figure 3*) or in vivo (*Figure 5*) of altering the number of MGE-derived cortical interneurons. These considerations led us to conclude that the DMSO and bumetanide alone control groups could be omitted without compromising our ability to determine whether maternal bumetanide treatment dose-dependently mitigates binge-type ethanol-induced aberrant tangential migration.

## Treatment of binge ethanol consuming dams with bumetanide prevents interneuronopathy in the PFC of young adult offspring

To determine the long-term effect of NKCC1 antagonism on ethanol-induced interneuronopathy, we quantified the distribution of PV$^+$ interneurons in the PFC of young adult mice that had been exposed prenatally to binge-type ethanol without or with bumetanide co-treatment (daily maternal i.p. injection, 0.15mg/kg, E13.5 - E16.5). The PFC was divided into subregions (anterior cingulate cortex (ACC), prelimbic (PL), infralimbic (IL)), and subdivided by cortical layer, based on the cytoarchitecture as revealed by DAPI counterstaining. Previously, we reported a layer V-specific increase in PV$^+$ interneurons in the young adult mPFC following binge-type ethanol exposure in utero (*Skorput et al., 2015*). In comparison, the number of PV$^+$ interneurons in the PFC of binge-type ethanol-exposed offspring treated in utero with bumetanide was significantly lower in layer V of the ACC (*Figure 5a and b*; EtOH $\bar{x}$ = 47.7 ± 3.5, 9 brains from 4 litters; EtOH+Bumet $\bar{x}$ = 37.3 ± 2.0, 4 brains from 2 litters; P = 0.00429; effect size ω$^2$ for layer = 0.79; effect size ω$^2$ for treatment = 0.02) and the PL region (*Figure 5a and c*; EtOH $\bar{x}$ = 30.9 ± 3.0, 9 brains from 4 litters; EtOH+Bumet $\bar{x}$ = 19.1 ± 1.5, 4 brains from 2 litters; P = 3.27x10$^{-5}$; effect size ω$^2$ for layer = 0.62; effect size ω$^2$ for treatment = 0.08). The increase in PV$^+$ interneurons of the ACC induced by in utero ethanol exposure was completely prevented by bumetanide co-treatment; there was no significant difference in the number of PV$^+$ interneurons in layer V of the ACC between offspring exposed to ethanol in utero that received maternal bumetanide treatment and controls (*Figure 5a and b*; Control $\bar{x}$ = 31.8 ± 2.0, 11 brains from 4 litters; P = 0.384). Maternal bumetanide treatment also completely prevented the ethanol-induced effect in layer V of the PL region (*Figure 5a and c*; Control $\bar{x}$ = 19.7 ± 1.9, 11 brains from 4 litters; P > 0.999). The effect of bumetanide treatment on preventing in utero ethanol-induced interneuronopathy extended throughout layer V of the PFC, with no significant difference in the number of PV$^+$ interneurons in the IL region compared to controls (*Figure 5a and d*; Control $\bar{x}$ = 8.4 ± 0.7, 11 brains from 4 litters; EtOH+Bumet $\bar{x}$ = 10.4 ± 2.1, 4 brains from 2 litters; P = 0.228; effect size ω$^2$ for layer = 0.65; effect size ω$^2$ for treatment = 0.07). Additionally, maternal treatment with bumetanide (0.15 mg/kg i.p. daily, E14.5 - E16.5) in the absence of in utero binge-type ethanol exposure (control liquid diet) did not alter the number of PV$^+$ interneurons in any of the PFC regional layers analyzed in young adult offspring (*Figure 5a - b*; Bumet, 3 brains from 2 litters; P > 0.247).

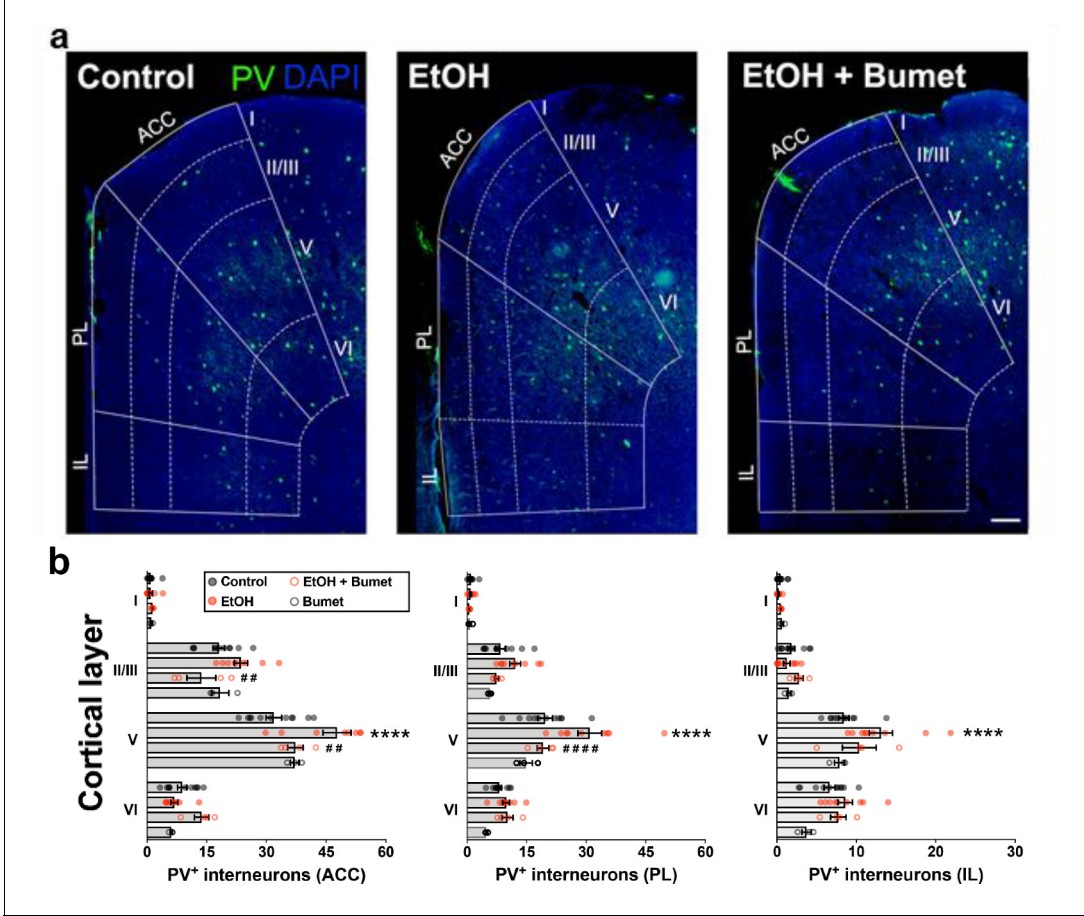

**Figure 5.** Treatment of binge ethanol-exposed dams with bumetanide prevents the interneuronopathy associated with ethanol exposure in utero. (a) Histological sections of young adult mouse prefrontal cortex processed for parvalbumin immunofluorescence, counterstained with DAPI, and binned by functional region and layer using DAPI cytoarchitecture. Scale bar = 100 μm. (b) Quantification of PV$^+$ MGE-derived GABAergic cortical interneurons in mice without (control), and with (EtOH) binge-type ethanol exposure (E13.5 - E16.5), as well as mice with ethanol exposure born to dams treated with bumetanide (0.15 mg/kg/day i.p. in DMSO; EtOH+Bumet), or mice born to dams that consumed control diet and received bumetanide treatment (Bumet). **** = p < 0.0001 compared to control; ##, #### = p < 0.01, 0.0001 compared to EtOH; two-way ANOVA with Bonferroni post-test analyzed per region, and stratified by treatment and cortical layer. ACC = anterior cingulate cortex, PL = prelimbic, IL = infralimbic. The EtOH group, and a portion of the Control group are reproduced from *Skorput et al. (2015)*.
DOI: https://doi.org/10.7554/eLife.48648.010

## Maternal bumetanide treatment prevents the deficits in behavioral flexibility seen with ethanol exposure in utero

To determine if the observed normalizing effect of maternal bumetanide treatment on tangential migration also mitigates deficits in PFC-dependent behavioral flexibility, we compared the performance of offspring exposed in utero to binge-type ethanol without and with maternal bumetanide treatment (0.15 mg/kg i.p, dissolved in DMSO) on the modified Barnes maze (*Figure 6*). Young adult offspring exposed to binge-type ethanol in utero and treated maternally with bumetanide exhibited no significant difference in the number of errors committed during the training and testing phases compared with ethanol-exposed, and control, offspring (*Figure 6a*; Control 14 offspring from four litters; EtOH 19 offspring from four litters; EtOH+Bumet 8 offspring from two litters; p>0.427).

As previously reported (*Skorput et al., 2015*), the errors committed by ethanol-exposed offspring increased on day one of the reversal phase (*Figure 6a*; Control $\bar{x}$ = 2.68 ± 0.37 errors; EtOH $\bar{x}$ = 4.78 ± 0.41 errors; P = 0.00192; effect size $\omega^2$ for trial = 0.18; effect size $\omega^2$ for treatment = 0.14). Offspring born to ethanol consuming dams treated with bumetanide, however, showed no significant increase in errors compared to controls (*Figure 6a*; EtOH+Bumet i.p. $\bar{x}$ = 3.28 ± 0.51 errors;

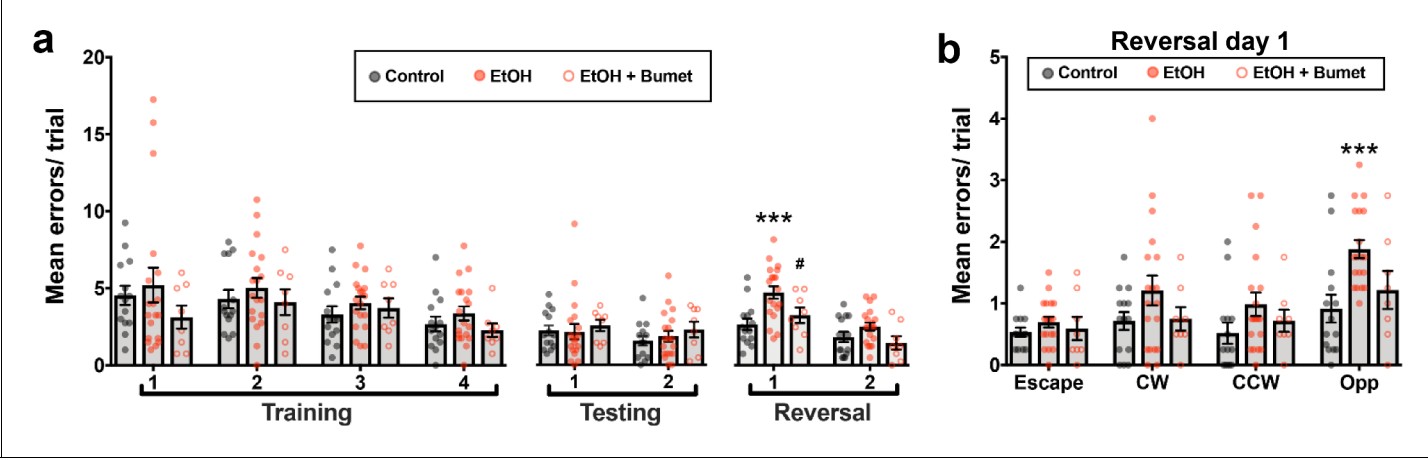

**Figure 6.** Maternal bumetanide treatment prevents the deficits in behavioral flexibility seen with ethanol exposure in utero. (**a**) Mean number of errors committed in the modified Barnes maze by young adult mice born to control, ethanol consuming (EtOH) and ethanol consuming plus bumetanide treated (EtOH+Bumet) dams across training testing and reversal phases of testing. *** = p < 0.001 compared to control, # = p < 0.05 compared to EtOH; two-way ANOVA with Bonferroni post-test. (**b**) Mean number of errors committed during the first reversal day stratified by quadrant relative to the escape hole. Increased errors in the opposite quadrant denotes perseverative behavior *** = p < 0.001 compared to control; two-way ANOVA with Bonferroni post-test. The Control and EtOH groups are reproduced from *Skorput et al. (2015)*.

DOI: https://doi.org/10.7554/eLife.48648.011

P >0.999), and was significantly lower than those committed by ethanol-exposed offspring not treated maternally with bumetanide (*Figure 6b*; P = 0.0433; effect size $\omega^2$ for quadrant = 0.11; effect size $\omega^2$ for treatment = 0.10). Further analysis revealed that the increase in errors exhibited by ethanol exposed offspring on day one of the reversal phase were committed in the opposite quadrant (*Figure 6b*; Control $\bar{x}$ = 0.92 ± 0.23 errors; EtOH $\bar{x}$ = 1.88 ± 0.15 errors; P = 0.00036). In contrast, the number of these perseverative errors committed by ethanol exposed offspring maternally treated with bumetanide was not significantly different from controls (*Figure 6b*; EtOH+Bumet $\bar{x}$ = 1.21 ± 0.31 errors; P = 0.0745). Thus, maternal bumetanide treatment prevented perseverative behaviors, which are indicative of reduced behavioral flexibility, in offspring exposed to ethanol in utero.

## Discussion

The results reported here advance our understanding of depolarizing GABAergic influence on tangential migration, and demonstrate that treatment of ethanol consuming dams with bumetanide can prevent persistent interneuronopathy in ethanol-exposed offspring. These results offer insights into the developmental pathophysiology of FASD, and posit therapeutic avenues for novel treatments.

As shown by our electrophysiological studies, ethanol-induced depolarizing shifts in $E_{GABA}$ of similar magnitude without and with bumetanide pretreatment, suggesting that the depolarizing shift of $E_{GABA}$ induced by ethanol occurs independent of NKCC1 function. Further, since NKCC1 antagonism alone significantly shifted $E_{GABA}$ to more hyperpolarized values, it is likely that bumetanide does not directly inhibit the ethanol-induced depolarizing shift in $E_{GABA}$, but rather normalizes it via a mechanism separate from NKCC1. Future studies examining the mechanism underlying the ethanol-induced shift in $E_{GABA}$ will be informative. It is plausible that ethanol shifts $E_{GABA}$ by enhancing the function of Cl$^-$ importers other than NKCC1, such as the sodium-independent anion exchange protein AE3. AE3 is expressed in the brain by E11 and augments the electrochemical gradient of chloride and bicarbonate in neurons, both of which are permeable through the GABA$_A$R (reviewed in *Hübner and Holthoff, 2013*). Importantly, this is likely an age-dependent phenomenon since GABA is depolarizing only during early development.

In this study, the normalization of [Cl$^-$]$_i$, and thus the chloride driving force, had the predicted effect of attenuating GABA induced depolarizing currents in migrating cortical interneurons. Previous work by *Cuzon et al. (2006)* and *Cuzon et al. (2008)* investigating the cell intrinsic and extrinsic

factors that contribute to GABA's influence on tangential migration argue strongly for $GABA_AR$-activated membrane depolarization influencing both prototypical migration of GABAergic interneurons and their abnormal migration induced by ethanol exposure (*Cuzon et al., 2008*; *Cuzon et al., 2006*). The importance of $GABA_AR$ signaling in tangential migration is consistent with work by others describing the dependence of interneuron migration on depolarization induced calcium signaling (*Behar et al., 1996*). Future investigations will need to address whether the binge paradigm used here alters extrinsic GABA influences, such as ambient GABA levels, and whether ethanol-induced potentiation of depolarizing GABA responses triggers activation of voltage-gated calcium channels and/or intracellular signaling cascades to induce aberrant migration of GABAergic interneurons.

The concept of a GABA-induced depolarization is not without controversy, as a number of studies suggest that this may be due to artifacts associated with in vitro experimental preparations that result in cellular injury and/or altered energy metabolism. Nonetheless, many other studies support the notion that GABA is depolarizing during development, and this controversy has been extensively reviewed (*Zilberter, 2016*). Overall, the results of our gramicidin perforated patch clamp experiments show that GABA exerts a depolarizing response in migrating MGE-derived GABAergic interneurons at embryonic ages, and that ethanol potentiates this response.

The ability of bumetanide to prevent aberrant tangential migration following in utero binge-type exposure to ethanol at the height of interneuron migration sheds light on the mechanistic basis of ethanol's enhancement of tangential migration. Maternal injection with a similar dose of bumetanide has been shown to shift the reversal potential of GABA mediated currents in neonatal cortical neurons, suggesting that maternally administered bumetanide reaches the embryonic brain (*Wang and Kriegstein, 2011*). With the doses of bumetanide used here, we did not see a reduction in tangential migration below levels seen in control animals, suggesting the existence of a therapeutic window in which bumetanide treatment can inhibit the aberrant migration induced by ethanol exposure in utero while not significantly effecting endogenous migration. Despite not impacting tangential migration in vitro, bumetanide at 20 µM did significantly shift the $E_{GABA}$ of migrating interneurons to more hyperpolarized values (*Figure 1e*). This discrepancy may be due to the limited magnitude of the shift in $E_{GABA}$, and/or a difference in the acute vs persistent effects of bumetanide on $E_{GABA}$ in MGE-derived GABAergic cortical interneurons. Further mechanistic investigations are warranted regarding the interactions of ethanol and bumetanide in augmenting $E_{GABA}$ in both acute, and chronic settings across a range of concentrations.

Our experiments testing the long-term outcome of bumetanide treatment during binge-type ethanol exposure at the height of interneuron migration offer insights into the role of interneuronopathy in the pathophysiology of FASD, and the potential for bumetanide as a preventative treatment. The prevention of in utero ethanol exposure-induced changes in interneuron migration and final positioning remain correlative with the observed normalization of behavioral flexibility. However, this work strongly implicates interneuronopathy in the etiology and symptomatology of FASD. Parvalbumin-expressing GABAergic interneurons in the mPFC pace gamma rhythms required for higher cognitive processing (*Kim et al., 2016*). These rhythms are disrupted by either subnormal or supernormal parvalbumin interneuron function (*Carlén et al., 2012*; *Sohal et al., 2009*). Thus, cognitive behavioral deficits resulting from I/E imbalance within intracortical circuits can become manifest regardless of the directionality of this skewing, for review see *Ferguson and Gao (2018a)*. To this end, it has been demonstrated that increased excitability among mPFC PV neurons results in increased perseverative behaviors (*Ferguson and Gao, 2018b*), consistent with the increased perseverative behavior reported here in mice with increased numbers of parvalbumin-expressing interneurons in the mPFC following prenatal ethanol exposure. It is unlikely that aberrant tangential migration is the only effect of ethanol that contributes to the imbalance of synaptic inhibition and excitation in FASD (*Kroener et al., 2012*; *Skorput et al., 2015*). However, bumetanide's ability to prevent this migratory defect in vivo suggests that such treatment may prevent ethanol-induced teratogenic effects on other cell populations that occur via potentiation of $GABA_AR$ activation. GABAergic interneuron migration appears to be sensitive to concentrations of ethanol that do not induce the widespread neuronal loss most commonly seen in the setting of high level exposure at a developmental time equivalent to late human gestation (*Farber et al., 2010*). Therefore, interneuronopathy may be an underlying etiology in a significantly larger population of FASD patients than is neurodegeneration. Dams treated with bumetanide and/or ethanol for the limited duration modeled here did not exhibit

significant deviation of body weight or physical condition compared to controls. However, the influence of maternal bumetanide effects on the endpoints measured here in offspring warrant future investigation. A potential caveat of this work is that the behavioral studies lacked the DMSO alone and bumetanide alone treatment groups. These groups were included in subsequent experiments that quantified interneuronopathy (*Figures 3* and *5*), and the results confirmed that DMSO or bumetanide, by themselves, have no effect on the ethanol-induced enhancement of interneuronal migration. The concentration of DMSO injected i.p. was 0.075%; the final concentration in vivo would have been expected to have been significantly lower. Future experiments investigating the therapeutic window of maternal bumetanide treatment would be enhanced by including these control groups. We have demonstrated a reduction in the perseverative behavior of mice exposed to ethanol in utero. However, other FASD cognitive phenotypes remain to be examined. In this light, future investigations should also assess other FASD-related behaviors and the cellular underpinnings in the associated cortical regions.

Preliminary studies report changes in fMRI activation and improvement in executive functioning with behavioral therapy (*Nash et al., 2017*). However, no treatments are currently available to ameliorate the developmental etiology of prenatal alcohol exposure. The work presented here offers antagonism of NKCC1 as a mechanism by which the short-term effects of binge-type ethanol exposure are mitigated, leading to prevention of a previously demonstrated ethanol-induced cognitive deficit. Indeed, developmental bumetanide exposure has been proposed for the treatment of neonatal epilepsy, and possible adverse effects have been reviewed (*Ben-Ari, 2012*; *Ben-Ari and Tyzio, 2011*). Our work shows that bumetanide has a biological effect on ethanol-enhanced tangential migration when administered maternally via both intraperitoneal and oral routes. While the co-administration of ethanol and bumetanide employed here served to assess mechanistic questions regarding GABAergic interneuron migration and the implications of interneuronopathy in FASD, the more likely clinical scenario would be administration of bumetanide after ethanol exposure (e.g. a patient learns she is 6 weeks pregnant, and reports binge drinking within that period of time). Therefore, the next step for this work is clearly to assess bumetanide's ability to prevent or mitigate the long-term deleterious effects of in utero ethanol exposure when administered at times after ethanol exposure.

While it is likely that some enhancement of tangential migration has occurred initially, it is possible that bumetanide treatment will slow migration of subsequent interneurons to the cortex. Alternatively, it may be that bumetanide treatment later in development, when cortical neurons have already reached their final location, may allow changes in synaptic integration to prevent consolidation of what would otherwise be an imbalanced circuit. Physiological maturation of PV$^+$ cortical interneurons closes the critical period of synaptic plasticity during corticogenesis (*van Versendaal and Levelt, 2016*). This maturation is driven by depolarizing GABAergic activity, and thus, is sensitive to the activity level of NKCC1 (*Deidda et al., 2015*; *van Versendaal and Levelt, 2016*). Indeed, work in the visual cortex demonstrates the ability for bumetanide treatment to exert lasting effects on intracortical circuit formation by extending the critical period of synaptic plasticity via a delay in the physiological maturation of PV$^+$ interneurons (*Deidda et al., 2015*). NKCC1 antagonist treatment may therefore be beneficial for mitigating interneuronopathy directly via a delay of MGE-derived interneurons, and/or indirectly by mitigating the deleterious effects of interneuronopathy on consolidation of intracortical circuits.

# Materials and methods

### Key resources table

| Reagent type (species) or resource | Designation | Source or reference | Identifiers | Additional information |
|---|---|---|---|---|
| Genetic reagent (*M. musculus*) | Nkx2.1-Cre | The Jackson Laboratory | JAX: 00861 MGI: J:131144 | PMID: 17990269 |
| Genetic reagent (*M. musculus*) | Ai14 | The Jackson Laboratory | JAX: 007914 MGI: J:155793 | PMID: 20023653 |

*Continued on next page*

Continued

| Reagent type (species) or resource | Designation | Source or reference | Identifiers | Additional information |
|---|---|---|---|---|
| Antibody | Anti-parvalbumin (mouse polyclonal) | Millipore Sigma | Cat#: MAB 1572 | 1:1000 |
| Chemical compound, drug | GABA | Millipore Sigma | Cat#: A2129 | |
| Chemical compound, drug | Bumetanide | Millipore Sigma | Cat#: B-3023 | |
| Software, algorithm | ImageJ | NIH | https://imagej.nih.gov/ij/ | |
| Software, algorithm | Adobe Photoshop | Adobe Systems, San Jose, CA | http://www.adobe.com/products/photoshop.html | |
| Software, algorithm | Graphpad Prism | Graphpad Software, Inc, La Jolla, CA | http://www.graphpad.com/ | |
| Software, algorithm | G*Power 3.1 | Heinrich Heine University Düsseldorf | RRID: SCR_013726 | |

## Animals

All procedures were performed in accordance with the National Institutes of Health *Guide for the Care and Use of Laboratory Animals* and approved by the Dartmouth Institutional Animal Care and Use Committee. The Nkx2.1-Cre transgenic mouse line (originally obtained from Dr. Stewart Anderson; (*Xu et al., 2008*) was crossed with the Ai14 Cre-reporter mouse line (Jackson laboratories, Bar Harbor, ME) to yield Nkx2.1Cre/Ai14 mice harboring tdTomato-fluorescent (Nkx2.1$^+$) MGE-derived GABAergic interneurons (*Skorput et al., 2015*). For time-pregnant mating, pairs of male and female mice were housed overnight, with the following day designated as E0.5, and the day of birth designated as postnatal day (P) 0. Embryonic day 13.5–16.5 embryos and P58-85 young adult mice of either sex were included in this study as available. No difference in the ratio of male to female offspring was noted between treatment groups. Embryonic day 13.5–16.5 and P58 - 85 were operationally defined to be within the age range equivalent to mid-first trimester (*Clancy et al., 2001*) and young adulthood (*Varlinskaya and Spear, 2004*), respectively in humans.

## Perforated patch clamp recording and drug application in acute embryonic telencephalic slices

### Acute slice preparation

On E14.5–E16.5, dams were asphyxiated with $CO_2$ and fetuses were removed by caesarian section. Nkx2.1Cre/Ai14 embryos were phenotyped by the presence of tdTomato fluorescence over the cortical region of the dissected brains, which is readily visualized using UV goggles. The brains expressing tdTomato fluorescence were isolated and immersed in ice-cold oxygenated artificial cerebral spinal fluid (aCSF) containing (in mM): NaCl 124; KCl 5.0; $MgCl_2$ 2.0; $CaCl_2$ 2.0; glycine 0.01; $NaH_2PO_4$ 1.25; $NaHCO_3$ 26; glucose 10. The brains were then embedded in 3.5% low-melting point agarose (Invitrogen, Carlsbad, CA), and coronal telencephalic slices (250 µm) were sectioned on a vibratome (Electron Microscopy Services, Hatfield, PA). The slices were stored immersed in a reservoir of aCSF at room temperature for approximately 1 hr prior to use for electrophysiological experiments. For consistency, we used only slices in which the MGE and LGE were clearly demarcated by the ganglionic sulcus.

### Perforated patch clamp recording

Gramicidin perforated patch clamp recording was used in order to preserve the intracellular chloride (Cl$^-$) concentration (*Ebihara et al., 1995*). An acute 200 µm telencephalic slice obtained from E14.5-E16.5 Nkx2.1Cre/Ai14 brain was transferred to a recording chamber, stabilized by an overlaying platinum ring strung with plastic threads, and maintained at 32–34° C on a heated stage fit onto a fixed-stage upright microscope (BX51WI, Olympus, Melville, NY). The slices were perfused at a rate of 0.5–1.0 ml/min with oxygenated aCSF containing (in mM): 125 NaCl, 2.5 KCl, 1 $MgCl_2$, 1.25 $NaH_2PO_4$, 2 $CaCl_2 \cdot 2H_2O$, 25 $NaHCO_3$, 25 D-glucose, pH 7.4 (adjusted with 1N NaOH). Under

fluorescence illumination and Hoffman Modulation Optics (Modulation Optics, Greenvale, NY), tdTomato fluorescent cells (Nkx2.1$^+$) were identified using a 40X water immersion objective (3 mm working distance; Olympus; *Figure 1a*). Real-time images were captured using an analog video camera attached to a video frame grabber (Integral Technologies, Indianapolis, IN) and displayed on a computer monitor, which also aided the navigation and placement of the recording and drug pipettes.

Perforated patch-clamp recording pipettes were pulled from borosilicate glass capillaries (1.5 mm outer diameter, 0.86 mm inner diameter; Sutter Instrument Co., Novato, CA). They were first front-loaded with a K-gluconate-based internal solution containing (in mM): 100 K-gluconate, 2 MgCl$_2$, 1 CaCl$_2$, 11 EGTA, 10 HEPES, 30 KCl, 3 Mg$^{+2}$ ATP, 3 Na$^+$ GTP (adjusted to pH 7.3 with 1N KOH) and then back-filled with the same solution supplemented with 10 μg/ml gramicidin. When filled with recording solution, the patch pipettes had resistances of 8–10 MΩ. Series resistance, monitored periodically throughout the recording, typically dropped within 10 min and stabilized for a sufficient length of time (~15 min) for the perforated patch recording experiments. The stability of the zero current baseline was also monitored continuously, and cells with unstable recordings were excluded from analysis. Recordings were made using an AxoPatch 700B amplifier (Molecular Devices Inc, Sunnyvale, CA). Membrane currents were digitized at 20 kHz (Digidata 1320A; Molecular Devices), recorded with low-pass filtering at 10 kHz (Digidata1320A; Molecular Devices Inc) and analyzed offline using GraphPad Prism software (Version 7.0).

## Drug application

GABA (MilliporeSigma, Burlington, MA) was dissolved in aCSF, stored as frozen stock and diluted to a working concentration of 50 μM with aCSF immediately prior to each recording session. Ethanol was prepared fresh by diluting 95% ethanol with aCSF to 6.5 mM. We used this low concentration of ethanol for the electrophysiological studies because it approximates the blood alcohol concentration (30 mg/dL) attained in a mouse model of a moderate to low level of chronic ethanol consumption used previously to investigate cortical development and function following prenatal ethanol exposure (*Cuzon et al., 2008*; *Skorput and Yeh, 2016*). Bumetanide (MilliporeSigma) was made into a salt to increase its solubility in water. A solution of sodium hydroxide was added to a suspension of bumetanide, the resultant suspension was heated to 70–80°C with stirring until a clear homogenate solution was obtained, which was then concentrated and dried to yield a white solid bumetanide sodium salt (3-butylamino-4-phenoxy-5-sulfamoyl-benzoic acid + Na; courtesy of Dr. Alex Pletnev, Department of Chemistry, Dartmouth College) that increased the solubility of bumetanide in water to 9 mg/ml. This bumetanide salt (MW = 362.42 g/mol) was dissolved either in aCSF for the experiments involving perforated patch recording (20 μM) or in liquid food in experiments that called for bumetanide being administered via the oral route to pregnant dams.

Drug solutions were loaded into separate barrels of a six-barrel drug pipette assembly that was navigated to within 10 μm of the soma of the cell under study and applied using regulated pulses of pressure (≤3 p.s.i.; Picospritzer, General Valve Corporation, Fairfield, NJ) (*Figure 1a*). The timing and the duration of the pressure pulses were controlled by a digital multi-channel timing unit and pulse generator (Pulsemaster A300, WPI, Sarasota, FL). One of the barrels of the multi-barrel assembly was routinely filled with aCSF, which was applied between drug applications to clear drugs from the vicinity of the cell and to control for mechanical artifacts that can occur occasionally due to bulk flow.

## Organotypic embryonic slice cultures

Time-pregnant female mice were sacrificed by carbon dioxide asphyxiation at E14.5. Embryos were removed, and the brains were harvested and processed as described previously (*Cuzon et al., 2008*). Briefly, the embryos were decapitated, the brains were isolated with aid of a dissecting microscope, and immersed in ice-cold slicing medium (1:1 F12:DMEM) oxygenated by bubbling with 95% O$_2$5% CO$_2$. Brains were then embedded in 3.5% low melting point agarose in 1:1 F12:DMEM. Using a vibrating microtome (Electron Microscopy Sciences), coronal slices (200 μm) were collected into ice-cold slicing media saturated with 95% O$_2$/5% CO$_2$.

The embryonic organotypic slices were maintained on a fine nylon mesh supported on a mid-gauged U-shaped platinum wire in a 35 mm round Petri dish. A small volume of sterile filtered culture media (0.8 ml) (1:1 F12/DMEM, 1% penicillin/streptomycin, 1.2% 6 mg/ml glucose in DMEM,

10% fetal bovine serum, and 1% L-glutamine) was added to achieve air-liquid interface. Slice cultures were placed in a humidified incubator (37°C, 5% $CO_2$) for 1 hr, after which the following culture media were prepared and used to replenish sister slice cultures every 6 hr over the next 24 hr: (1) 50 mM EtOH; (2) 20 µM bumetanide; (3) 50 mM EtOH + 20 µM bumetanide; (4) vehicle (DMSO, 7.5 × $10^{-4}$%). The concentration of ethanol (50 mM) was relatively high but well within the conventional range of concentrations (up to 100 mM) employed in studies involving ethanol exposure in vitro (*Kumada et al., 2006*; *Larsen et al., 2016*; *Xiang et al., 2015*; *Zou et al., 1993*). As detailed in the Results section, 50 mM ethanol yielded similar outcomes in terms of ethanol's effect on migration, shift in $E_{GABA}$ and potentiation of responses to GABA.

Following 24 hr in culture, the slice cultures were washed in PBS and immerse-fixed overnight in 4% PFA/0.1 M PBS at 4°C. Following cryoprotection in 30% sucrose/0.1M PBS, the slice cultures were resectioned at 30 µm using a sliding microtome, collected in PBS, mounted on charged slides, counterstained with DAPI and coverslipped with FluorSave Reagent (Calbiochem, La Jolla, CA).

## Binge-type maternal ethanol consumption and in vivo bumetanide administration

In the in vivo experiments, ethanol exposure in utero was begun on E13.5 and terminated on E16.5 in order to be within the timeframe when tangential migration of MGE-derived cortical interneuron is at its peak in mouse (*Anderson et al., 2001*; *Batista-Brito and Fishell, 2009*; *Gelman et al., 2009*; *Hladnik et al., 2014*; *Jiménez et al., 2002*; *Marín and Rubenstein, 2001*; *Parnavelas, 2000*). Pregnant dams were individually housed and assigned to one of two groups: ethanol-fed, or control-fed. Mice were maintained under normal 12/12 hr light/dark cycle on a liquid diet (Research Diets, New Brunswick, NJ) supplemented with ethanol (5% w/w; ethanol-fed group) or an isocaloric control diet containing maltose (control-fed group); water was available ad libitum. The liquid food was replenished daily between 3:00 and 5:00 PM, when the amount consumed was measured and the dams weighed. Mice were maintained on their respective diets from E13.5 until E16.5, after which they were returned to standard chow. Dam blood alcohol level (BAL; 80 ± 21 mg/dl) was assessed using an Analox Instruments GM7 series analyzer (Lunenburg, MA), with blood collected via the tail vein at 11:30 PM on E15.5. Dams subjected to our regimen of binge-type gestational ethanol consumption carried their offspring to full term, litter size was unaffected (7.50 ± 0.62 for Control pups; 7.67 ± 0.76 for EtOH-exposed pups; unpaired t-test p>0.05), and offspring in all control and experimental cohorts developed normally to young adulthood for behavioral testing (*Skorput et al., 2015*).

Bumetanide solution for intraperitoneal (i.p.) injection was prepared fresh from powder at the start of each binge epoch by adding bumetanide to normal saline at 0.015 mg/ml or by dissolving bumetanide in DMSO at 20 mg/ml stock concentration that was diluted to 0.015 mg/ml in normal saline for daily i.p. injections. Bumetanide was injected into binge ethanol-consuming dams at 10 ml/kg, yielding a dose of 0.15 mg/kg/day over the 3-day binge period. Normal saline or bumetanide i.p. injections did not alter the amount of liquid food consumed (one-way ANOVA; p = 0.62) or the BAL (one-way ANOVA; p = 0.99) of dams fed with ethanol. Bumetanide delivery via liquid food was accomplished by dissolving the bumetanide sodium salt in the water used to make 5% ethanol-containing liquid food at; 1.3, 13, 130, 320 mg of bumetanide salt per kg of food.

## Imaging and analysis of immunofluorescence and Nkx2.1$^+$ MGE-derived GABAergic interneurons in the prefrontal cortex

Time-pregnant dams were euthanized by $CO_2$ asphyxiation on E16.5. The embryos were quickly removed, their brains dissected, immerse-fixed overnight at 4°C in PBS containing 4% paraformaldehyde (PFA)/0.1M phosphate buffered saline (PBS), and then cryoprotected in 30% sucrose/0.1M PBS. Cryosections (30 µm) were cut with a sliding microtome, mounted on glass slides, DAPI counterstained and cover slipped with FluorSave Reagent (Calbiochem, La Jolla, CA). The embryonic prefrontal cortex (PFC) was delineated as part of the dorsomedial telencephalon based on DAPI counterstaining of the same sections used for counting and analyzing cells. For each embryonic brain, 10 consecutive sections of the embryonic PFC beginning at equivalent rostral-caudal levels were analyzed for counts of Nkx2.1$^+$ cells.

Young adult mice were perfused trans-cardially with cold PBS and then with 4% PFA/0.1 M PBS. Brains were removed and immerse fixed in 4% PFA/0.1M PBS overnight at 4℃. Following cryoprotection in 30% sucrose, 30 μm coronal sections were made on a sliding microtome, and collected into PBS. Young adult tissue sections were blocked for 1 hr at room temperature in PBS containing 10% NGS and 0.05% Triton X-100. Then incubated overnight at 4° C with mouse anti-parvalbumin primary antibody (Millipore) at a dilution of 1:1000 in PBS containing 1% NGS and 0.01% Triton X-100. Following 3X washing in PBS, sections were incubated overnight with a 1:1000 dilution of Alexa Fluor 555 conjugated goat-anti-mouse secondary antibody (Invitrogen, Grand Island, NY) in PBS containing 1% NGS and 0.01% Triton X-100. Negative control with primary antibody omitted was routinely processed in parallel. For each young adult brain, sections were defined as containing the PFC beginning with the rostral most section in which all five layers of the PFC were visible, and extending caudally to the decussation of the corpus callosum. Within this operationally defined region of the PFC, images were captured from ten consecutive sections per animal beginning at equivalent rostral-caudal levels. Parvalbumin-immunopositive cells were manually counted per subdivision and per layer by trained experimenters blinded to the treatment conditions using Image J software.

Fluorescent images were captured digitally using a CCD camera (Hamamatsu, Hamamatsu city, Japan) fitted onto a spinning disk confocal microscope (BX61WI; Olympus, Melville, NY) and controlled by IPLab 4.0 software (BD Biosciences, San Jose, CA). Images were stitched using Fiji image J (*Preibisch et al., 2009*) to yield a full view of the region of interest. For in vivo experiments, counting of PV$^+$ or Nkx2.1$^+$ MGE-derived GABAergic interneurons in the embryonic or young adult PFC was automated using Fiji's auto segmentation algorithm (RenyiEntropy) within the manually defined region of interest. For in vitro experiments, histological sections of organotypic slice cultures were imaged at 4X and the images were montaged to allow visualization of the cortex from the corticostriate junction to the dorsal apex. One hundred micron consecutive bins spanning the thickness of the cortex were organized along the ventral to dorsal extent of the cortex, with the starting point of the first bin aligned at the corticostriate juncture (*Cuzon et al., 2006*). The Nkx2.1$^+$ MGE-derived GABAergic interneurons within these bins were quantified by trained experimenters blinded to the experimental condition using Fiji's cell counting tool.

## Modified Barnes maze

Young adult mice (P58-70) were tested behaviorally using a modified Barnes maze as previously reported (*Koopmans et al., 2003*; *Skorput et al., 2015*). Briefly, the maze consisted of twelve equally-spaced holes around a circular wall (d = 95 cm), with spatial cues (large red letters) placed between the holes along the interior surface of the circular wall. Each mouse was randomly assigned an escape hole within a quadrant. All other holes were plugged. Mice were trained to find the escape hole with four 4 min trials per day for 4 consecutive days. Each trial ended when the mouse entered the escape hole. The mouse was then returned to its home cage (escape reinforcement). Following the training phase, the mice were rested for 2 days, and then tested for their ability to recall the position of the escape hole on the next 2 days, with four 4 min trials per day (testing phase). After 2 days of testing, the location of the escape hole was switched to the hole directly opposite its initial position (reversal phase). Mice were then tested for their ability to find the reversed escape hole with four 4 min trials per day over the subsequent 2 days.

At the beginning of each trial, the mouse was placed in the center of the maze and covered with a start box; time began when the box was removed. Latency to enter escape hole, time per quadrant; Home (H), Clockwise (CW), Opposite (Opp.), Counter Clockwise (CCW) (clock directions are relative to escape quadrant), and distance per quadrant measurements were made via video tracking software running an overhead camera hung at a set distance from the surface of the maze. During a trial, errors made (nose pokes in non-escape hole) and their locations, were recorded manually by observers on opposite sides of the arena.

## Statistics

All young adult histological data were acquired from the defined subregions and layers within the PFC. All groups consisted of data acquired from ten 30 μm tissue sections per animal (n = 1: 1 animal = 10 sections) from a minimum three individual animals of multiple litters. For

electrophysiological data, n refers to the number of litters used in order to minimize litter effects, and the total number of cells recorded for each experiment is noted in results. As the variability (standard deviation) of histological and behavioral end points was larger between individual offspring than that between data averaged per litter, we reported the mean data per individual offspring ± standard error. Reporting the more variable unit of determination is more representative of the true biological variability (*Brien et al., 2006*). All groups consisted of data acquired from a minimum three individual offspring from multiple litters. Power calculations were performed using the G*Power 3.1 software depending on whether the data were analyzed using t-test or ANOVA. Variance and expected differences were estimated by using group means and standard deviations from preliminary data or past experience in similar studies and reviews of related use in the literature. Group means were compared by unpaired t-test, one-way ANOVA or two-way ANOVA with appropriate post-hoc test as indicated, and reported in figure legends, and reported as mean ($\bar{x}$) ± standard error in the results section. Reported exact p values are multiplicity adjusted for ANOVA testing of multiple comparisons. The effect sizes reported (Cohen's d for t-tests and $\omega^2$ for ANOVAs) were calculated using the Graphpad Prism software and formulas from Lakens (2013). We note that previously published counts of prenatal and postnatal cortical interneurons and behavioral data obtained from control, and prenatal binge ethanol-exposed young adult mice (*Skorput et al., 2015*) were included for comparison with the bumetanide-treated cohort in the present study. This was justified as, in practice, the experiments involving all three cohorts were conducted in parallel, and reporting of the bumetanide data required the subsequent mechanistic studies reported here, specific instances of reproduction are outlined in the relevant figure legends.

## Acknowledgements

The authors acknowledge funding from the National Institute on Alcohol Abuse and Alcoholism at the National Institutes of Health PHS NIH R01 AA023410, AA027754, R21 AA024036 to HHY, and F31 AA027694-01 to SML.

## Additional information

### Funding

| Funder | Grant reference number | Author |
|---|---|---|
| National Institutes of Health | PHS NIH R01 AA023410 | Hermes H Yeh |
| National Institutes of Health | PHS NIH R21 AA024036 | Hermes H Yeh |
| National Institutes of Health | PHS NIH R01 AA027754 | Hermes H Yeh |
| National Institutes of Health | PHS NIH F31 AA027694-01 | Stephanie M Lee |

The funders had no role in study design, data collection and interpretation, or the decision to submit the work for publication.

### Author contributions

Alexander GJ Skorput, Conceptualization, Formal analysis, Validation, Investigation, Visualization, Methodology, Writing—original draft, Project administration, Writing—review and editing; Stephanie M Lee, Formal analysis, Validation, Investigation, Visualization, Methodology, Writing—original draft, Project administration, Writing—review and editing; Pamela WL Yeh, Resources, Project administration; Hermes H Yeh, Conceptualization, Resources, Supervision, Funding acquisition, Project administration, Writing—review and editing

### Author ORCIDs

Stephanie M Lee (iD) https://orcid.org/0000-0001-6597-7144
Hermes H Yeh (iD) https://orcid.org/0000-0001-6733-3692

## Ethics

Animal experimentation: All animal experimental procedures were performed in accordance with the National Institutes of Health Guide for the Care and Use of Laboratory Animals and approved by the Dartmouth Institutional Animal Care and Use Committee (IACUC Protocol# 00002109(m18a)).

## Decision letter and Author response

Decision letter https://doi.org/10.7554/eLife.48648.014
Author response https://doi.org/10.7554/eLife.48648.015

## Additional files

### Supplementary files
• Transparent reporting form
DOI: https://doi.org/10.7554/eLife.48648.012

### Data availability
All data generated or analyzed during this study are included in the manuscript.

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
