## [Decision Letter]

Thank you for submitting your article "The NKCC1 antagonist bumetanide mitigates interneuronopathy associated with ethanol exposure In utero" for consideration by *eLife*. Your article has been reviewed by three peer reviewers, and the evaluation has been overseen by Joseph Gleeson as the Reviewing Editor and Huda Zoghbi as the Senior Editor. The following individual involved in review of your submission has agreed to reveal their identity: C Fernando Valenzuela (Reviewer #1).

The reviewers have discussed the reviews with one another and the Reviewing Editor has drafted this decision to help you prepare a revised submission.

Summary:

The study of Skorput et al. investigated the effect of pharmacological antagonism of NKCC1 (with bumetanide) on ethanol-induced alterations of tangential interneuronal migration in the prefrontal cortex. This manuscript represents an extension of prior work from the same laboratory and other laboratories that showed that prenatal ethanol exposure increases migration of interneurons from the medial ganglionic eminence (MGE) likely through increased ambient GABA concentration and enhanced embryonic GABAergic signaling (Cuzon et al., 2008). In addition the Yeh laboratory has shown in prior work that gestational ethanol exposure caused (1) an enduring increase in the number of Parvalbumin+ interneurons in the medial prefrontal cortex, (2) increased spontaneous and evoked GABA-ergic synaptic transmission on layer V pyramidal neurons and (3) altered reversal behavior in a Barnes maze test (Skorput et al., 2015).

In this study, Skorput et al. confirm that EtOH potentiates the depolarizing actions of GABA on migrating (Nkx2.1+) interneurons in the mouse model of E13.5-16.5 binge-type ethanol exposure. The authors then show that bumetanide, an inhibitor of the NKCC1 chloride transporter, could counteract the effects ethanol's potentiation of GABAergic depolarization. Bumetanide was able block ethanol's enhancement of interneuron migration from the MGE both in slice preparation and in vivo. In addition bumetanide protected gestational ethanol – exposed embryos from a behavioral deficit in reversal behavior in the Barnes maze test. From this the authors argue that bumetanide may be potential therapeutic approach to ameliorating FASD. The significance of this study is twofold.

1) This study demonstrates that enhancement of depolarizing action of GABA is necessary for fetal ethanol-induced aberrant migration and the excess of interneurons in PFC, because abrogating the enhanced depolarizing action of GABA prevents the aberrant migration and the excess of interneurons in PFC by the ethanol exposure.

2) This study posits NKCC1 as a pharmacological target for the management of FASD.

While two of the reviewers were positive, one reviewer commented since many of the findings have been reported earlier, the significance of the work rests largely on the bumetanide reversal of the ethanol effect. While potentially interesting, the potential neuroprotective effects of bumetanide are documented in only one behavioral test and in one type of neurons in one area of cortex. Whether bumetanide is broadly protective for fetal alcohol exposure is unknown and thus the finding seems somewhat narrow for publication in *eLife*.

Essential revisions:

- For the studies with organotypic slice cultures, a substantially higher ethanol concentration was used (50 mM vs. 6.5 mM for the acute slice studies vs. 80 mg/dl (17 mM) for the in vivo studies) and the reason for doing this was not provided. Ethanol is known to exert dose-dependent effects that could be non-linear (inverted U shaped). Have the investigators tested if 50 mM affects E_GABA_ and GABA_A_ responses in the same direction as lower ethanol concentrations?

- In Figures 4-6, it is critical to measure the effect of bumetanide on blood ethanol concentrations in the dams. Bumetanide is a loop diuretic that could affect liquid diet or water consumption in the dams. The effects of bumetanide should be confirmed with a different approach (manipulation of NKCC1 expression?). In addition, the definition of the unit of determination for these studies was arbitrarily changed to an animal. Justification for doing this should be provided.

- Several important controls are missing in Figures 4 and 6. DMSO alone, bumetanide alone, etc. Please include these controls or explain why they were not performed.

- Discrepancy of the bumetanide effects on the control between the reversal potential E_GABA_ (Figure 1E, rightmost) and interneuron migration (Figure 3B, rightmost). This reviewer agrees with the authors' view, but is wondering if the authors have data on the GABA response amplitude (peak current amplitude) comparing aCSF and aCSF + bumetanide conditions, in the Figure 2 experiments.

- Relationship between the increased number of interneurons in PFC and the impaired behavioral flexibility requires clarification. In their previous paper (Skorput et al., 2015), the authors referred to the work by Cho et al. (2015, Neuron, 85:1332-1343). But the latter mouse showed reduced number of interneurons in the prefrontal cortex. To prove that the interneuronopathy of this type (increase in the number of interneurons) is the etiology of impaired behavioral flexibility, an additional experiment that increases the number of interneurons in the PFC by other means than ethanol, or previous reports on mutant mice with similar increase in the number of interneurons, should be considered, either discussed or performed. If not presented then greater caution should be exercised with regard to the proposed mechanism.

---

## [Author Response]

Essential revisions:- For the studies with organotypic slice cultures, a substantially higher ethanol concentration was used (50 mM vs. 6.5 mM for the acute slice studies vs. 80 mg/dl (17 mM) for the in vivo studies) and the reason for doing this was not provided. Ethanol is known to exert dose-dependent effects that could be non-linear (inverted U shaped). Have the investigators tested if 50 mM affects E_GABA_ and GABA_A_ responses in the same direction as lower ethanol concentrations?

The reviewer appropriately points out (1) that we neglected to address whether or not 50 mM ethanol affects E_GABA_ and GABA_A_ responses in the same direction as the lower ethanol concentrations because (2) ethanol can exert inverted U-shaped concentration-dependent effects (our lab was one among several that noted this observation in the 1990s).

We now include supplementary figures summarizing data derived from new experiments and indicating that 50 mM ethanol indeed shifts E_GABA_ in the same direction (Figure 1—figure supplement 1) and potentiates GABA-activated current responses (Figure 2—figure supplement 1) as do the lower concentrations of ethanol. Indeed, our earlier studies on ethanol-GABA interaction standardly used 50 mM because it was the concentration of ethanol that we found to be consistently most effective in potentiating GABA responses (peak of the inverted U-shaped concentration/response curve). Also for this reason, we chose to use 50 mM in our initial studies with the organotypic embryonic slices in order to see an effect. However, although many other studies have used ethanol concentrations up to 100 mM (e.g., Zou et al., 1993; Kumada et al., 2006), we realized that 50 mM is a high and non-physiological concentration of ethanol. Therefore, in subsequent studies, we lowered the ethanol concentration to 17 mM or 6.5 mM, which are equivalent to the blood alcohol concentrations that we obtain in our binge-type or chronic maternal ethanol consumption mouse models, respectively (and yielded the same results). We have added a comment to this effect in the Materials and methods section (subsection “Organotypic embryonic slice cultures”, second paragraph) and revised the Results section accordingly (subsection “Ethanol induces a depolarizing shift in the GABA reversal potential of embryonic MGE-derived GABAergic cortical interneurons that is normalized by the NKCC1 inhibitor bumetanide”, end of first paragraph; subsection “Bumetanide attenuates ethanol-induced potentiation of depolarizing GABA responses in embryonic MGE-derived GABAergic cortical interneurons”).

- In Figures 4-6, it is critical to measure the effect of bumetanide on blood ethanol concentrations in the dams. Bumetanide is a loop diuretic that could affect liquid diet or water consumption in the dams.

Dams injected i.p. with bumetanide consumed similar amounts of the liquid food compared to controls, and the blood ethanol concentrations were similar as well. A statement to this effect has been added in the Materials and methods section (subsection “Binge-type maternal ethanol consumption and in vivo bumetanide administration”, last paragraph).

The effects of bumetanide should be confirmed with a different approach (manipulation of NKCC1 expression?).

We agree that a different approach through manipulation of NKCC1 expression would have enhanced the significance and validated the role of NKCC1 in the ethanol-induced interneuronopathy. We looked into obtaining the constitutive NKCC1 knockout mouse line from the Jackson Laboratories (stock No: 34262-JAX), but 30% of the homozygotic knockout mice die by the time of weaning. In addition, they are 80% smaller than wildtype as adults, have difficulty maintaining their balance, are deaf and, thus, are not suitable for investigating the role of NKCC1 in tangential migration of GABAergic interneurons. We also attempted to acquire an NKCC1-floxed mouse line from Germany but found the material transfer conditions unacceptable and shipping logistics prohibitive. Therefore, in the current work, we resorted to using the pharmacological inhibitor of NKCC1, bumetanide.

In addition, the definition of the unit of determination for these studies was arbitrarily changed to an animal. Justification for doing this should be provided.

We thank the reviewer for pointing this out. This is now highlighted and justified in the “Statistics” subsection of the Materials and methods. In our analyses, we used the unit of determination that offered the largest degree of biological variability. For the histological and behavioral studies, the variability (standard deviation) among individual offspring was larger than that between data averaged per litter (consistent with previous studies modeling FASD and noting this difference in variability e.g., Brien et al., 2006). Therefore, we used individual offspring as the unit of determination as it is a better reflection of the true biological variability.

- Several important controls are missing in Figures 4 and 6. DMSO alone, bumetanide alone, etc. Please include these controls or explain why they were not performed.

Figure 4: In the experiment illustrated in Figure 4, we felt justified not to include DMSO alone administered i.p. as one of the control cohorts because bumetanide administration via a different route that did not use DMSO as solvent (i.e., bumetanide salt in food) produced similar results. In addition, bumetanide in normal saline delivered i.p. controlled for the i.p. route of administration and, as presented in Figure 3 (in vitro) and Figure 5 (in vivo), bumetanide alone did not alter interneuron number. These observations, in toto, led us to bypass DMSO and Bumetanide alone as control cohorts in Figure 4 in order to minimize the use of experimental animals while effectively testing the hypothesis that bumetanide inhibits ethanol-induced enhancement of MGE derived interneurons in vivo. A statement has been added to the Results section to highlight this caveat (subsection “Maternal bumetanide treatment prevents ethanol-induced escalation of tangential migration in vivo”, last paragraph).

Figure 6: The behavioral experiments were performed in parallel with the studies reported in Skorput et al. (2015). We acknowledge that the DMSO and bumetanide only cohorts were not included in our behavioral testing. However, we did control for DMSO and bumetanide alone in subsequent experiments that quantified interneuronopathy (Figures 3 and 5). A statement has been added to the sixth paragraph of the Discussion to highlight this caveat, as well as to address the scope of the behavioral findings, the latter raised as a concern by one reviewer.

- Discrepancy of the bumetanide effects on the control between the reversal potential E_GABA_ (Figure 1E, rightmost) and interneuron migration (Figure 3B, rightmost). This reviewer agrees with the authors' view, but is wondering if the authors have data on the GABA response amplitude (peak current amplitude) comparing aCSF and aCSF + bumetanide conditions, in the Figure 2 experiments.

Comparison of aCSF and aCSF + bumetanide revealed that the two are not significantly different (unpaired two-tailed t-test, p = 0.71). This is now stated in the Results section (subsection “Bumetanide attenuates ethanol-induced potentiation of depolarizing GABA responses in embryonic MGE-derived GABAergic cortical interneurons”), and a supplementary data set is provided (Figure 2—figure supplement 2).

- Relationship between the increased number of interneurons in PFC and the impaired behavioral flexibility requires clarification. In their previous paper (Skorput et al., 2015), the authors referred to the work by Cho et al. (2015, Neuron, 85:1332-1343). But the latter mouse showed reduced number of interneurons in the prefrontal cortex. To prove that the interneuronopathy of this type (increase in the number of interneurons) is the etiology of impaired behavioral flexibility, an additional experiment that increases the number of interneurons in the PFC by other means than ethanol, or previous reports on mutant mice with similar increase in the number of interneurons, should be considered, either discussed or performed. If not presented then greater caution should be exercised with regard to the proposed mechanism.

We thank the reviewer for pointing this out – it was our oversight not having discussed this important issue. Our view is that cognitive behavioral deficits resulting from I/E imbalance within intracortical circuits manifest regardless of the directionality of the skewing. In addition to the study cited in Skorput et al. (2015) and referenced by the reviewer (Cho et al., 2015), study of mouse cognition has shown that pharmacologic activation of the GABAergic system in the prefrontal cortex with the GABA_A_R agonist muscimol reduces behavioral flexibility (Rich and Shapiro, 2007). In addition, Ferguson and Gao (2018b) demonstrated that enhanced inhibitory tone in the mPFC induced by a pharmacogenetic increase in the excitability of PV neurons resulted in increased perseverative behaviors. These findings and others were recently reviewed by Ferguson and Gao (2018a) in which they also conclude that “Modulation of PV activity has also been useful in demonstrating one of the more obvious principles of E/I balance, that too much inhibition can also impair PFC circuit function”.We have added a statement in sixth paragraph of the Discussion section to address this important point.